# Cocrystallization of Progesterone with Nitrogen Heterocyclic Compounds: Synthesis, Characterization, Calculation and Property Evaluation

**DOI:** 10.3390/molecules28104242

**Published:** 2023-05-22

**Authors:** Juan Xu, Wei Gao, Qi Zhang, Lifeng Ning

**Affiliations:** 1NHC Key Laboratory of Reproductive Health Engineering Technology Research, National Research Institute for Family Planning, Beijing 100081, China; xujuan@nrifp.org.cn; 2School of Pharmacy, Guangdong Pharmaceutical University, Guangzhou 510006, China; gaowei@gdpu.edu.cn; 3School of Chemistry and Chemical Engineering, Beijing Institute of Technology, Beijing 100081, China

**Keywords:** progesterone, pharmaceutical cocrystals, nitrogen heterocyclic compounds, cocrystal particles with suitable size, metadynamics-genetic crossing

## Abstract

Progesterone injection is oily because of its poor solubility. It is necessary to develop new dosage forms or delivery methods for Progesterone. Six cocrystals of Progesterone with nitrogen heterocyclic compounds (2,6-diaminopyridine, isonicotinamide, 4-aminopyridine, aminopyrazine, picolinamide and pyrazinamide) have been designed and prepared by ethyl acetate-assisted grinding, of which four cocrystals (2,6-diaminopyridine, isonicotinamide, 4-aminopyridine and aminopyrazine) had single crystal data in 1:1 stoichiometry. Metadynamics-genetic crossing was used to search and optimize various cluster structures to explain the reason the other two cocrystals could not be obtained with suitable size for single crystal X-ray diffraction. In contrast to the carboxyl group, the amide group and amino group were good substituents in the pyridine/pyrazine ring for cocrystallization with Progesterone, which meant inductive effect played an important role in nitrogen heterocyclic compounds containing reactive hydrogen. All cocrystals were more soluble than Progesterone in water, and Progesterone–pyrazinamide cocystal featured the best water solubility performance with an approximately six-fold increase over free Progesterone. This successful attempt provides an effective route for designing and manufacturing novel solid states of Progesterone.

## 1. Introduction

Progesterone (PRO, Table 1) is the main bioactive progestational hormone secreted by the ovary [1]. PRO can not only induce the transition of the endometrium to the secretory stage and increase endometrium receptance to facilitate the implantation of a fertilized egg but also act on the uterus, providing a good internal environment for the maintenance of pregnancy [2,3]. PRO injection is oily because of its poor solubility [4,5]. The advantages of oily injection are the exact curative effect and low price, while the disadvantages are injection site pain, stimulation and scleroma. Due to the first-pass effect of the liver, the absolute bioavailability of oral PRO is only 6~8% [6,7]. It is necessary to develop new dosage forms or delivery methods of PRO.

A pharmaceutical cocrystal is a binary or multi-component crystal system formed by the combination of drug active ingredient (API) and cocrystal former (CCF) through hydrogen bonds or other weak interactions between molecules [8,9,10]. By selecting CCFs with different properties, the physicochemical properties of API can be improved or designed at the molecular level without changing its structure, which is very important to maintain the biological activity of API in vivo [11,12,13,14]. PRO is a good substrate for cocrystallization because there are two carbonyl groups in the molecule that can act as hydrogen bond acceptors, and the steroidal parent nucleus can produce conjugation effects [15,16,17,18].

With our consistent aim to further explore the solid-state forms of PRO, cocrystals of PRO with 2,6-diaminopyridine (DMP), isonicotinamide (INA), 4-aminopyridine (AP), aminopyrazine (APZ), picolinamide (PA) and pyrazinamide (PZA) were prepared and evaluated in stability and water solubility (Figure 1). Four cocrystals (DMP, INA, AP, APZ) had single crystal data in 1:1 stoichiometry. Meanwhile, two cocrystals (PA, PZA) had no suitable size for single-crystal X-ray diffraction. The reasons were calculated by MTD-GC (metadynamics-genetic crossing). The calculation results showed that the tetramers (2PRO/2PA) were stable, and the structural fluctuations of the tetramers cluster in ethyl acetate solvent were increased and led to a deficiency in order structure. In contrast to the carboxyl group, the amide group and amino group were good substituents in the pyridine/pyrazine ring. This meant that for the cocrystallization of nitrogen heterocyclic compounds containing reactive hydrogen, the inductive effect played an important role. The cocrystals have been characterized by nuclear magnetic resonance, infrared spectroscopy, thermogravimetric analysis, differential scanning calorimetry, scanning electron microscopy, powder X-ray diffraction and single crystal X-ray diffraction (for PRO-DMP, INA, AP, APZ). The stability and solubility have also been explored systematically.

## 2. Results

### 2.1. General Analysis

The PXRD patterns of six cocrystals showed that the diffraction peaks had obvious differences in the position, number, strength, geometric topology and so on (Figure 2). SEM images showed that each cocrystal had unique crystal habits and morphological characteristics. PRO had block morphology, PRO-DMP and PRO-PA had smaller flake morphology, PRO-INA had bigger flake morphology, PRO-APZ, PRO-AP and PRO-PZA had irregular morphology (Appendix A). In FT-IR spectra (Appendix A), the asymmetric stretching vibrations at 3500–3100 cm^−1^ indicated the presence of the amino or amide groups in all six cocrystals. The ^1^H-NMR spectra clearly showed the 1:1 stoichiometry of PRO to the CCFs in all six cocrystal forms (Appendix A). The chemical shift of PRO in ^1^H-NMR was between 0.67 and 5.63 ppm, and nitrogen heterocyclic compounds had a larger chemical shift between 5.63 and 9.18 ppm. A sharp endothermic peak at 132 °C was shown in the TGA-DSC curve of PRO, corresponding to the melting point of PRO. Six cocrystals had similar thermodynamic behaviors with weight loss (Appendix A). The melting points of PRO and six cocrystals are shown in Table 2.

### 2.2. Single X-ray Diffraction Experiments

The crystallographic parameters of PRO-DMP/INA/AP/APZ are listed in Table 3. The single X-ray diffraction pattern is shown in Figure 3. PRO-DMP/INA/AP/APZ cocrystals were composed of PRO and CCF with a ratio of 1:1.

### 2.3. Stability Analysis

Stability experiment was performed to investigate the physical stability and transformation of the sample under different storage environments and physiological conditions. PRO and six cocrystals all showed excellent physical stability in high temperature, high humidity, high light intensity condition and in water suspension conditions (Appendix A).

### 2.4. In Vitro Dissolution Tests 

In preparing new alternative crystal forms of chemical entities, water solubility was one of the most important physicochemical parameters. In addition, solubility was closely related to the bioavailability of drugs in vivo. Since PRO has poor solubility in water (0.41 mg·mL^−1^), the synthesis of cocrystals is attractive. PRO-DMP/INA/AP/APZ/PA featured 1.22 mg·mL^−1^, 1.74 mg·mL^−1^, 1.81 mg·mL^−1^, 1.55 mg·mL^−1^ and 1.63 mg·mL^−1^, respectively, being 3- to 4.5-fold equilibrium concentration in water when compared with PRO used in the solubility experiments (Figure 4). In particular, for PRO-PZA, the equilibrium concentration reached 2.65 mg·mL^−1^, which was approximately 6.5-fold as large as the solubility of free PRO. This result suggested that PRO-PZA could be a suitable candidate for novel PRO pharmaceutical formulations with improved solubility. Beyond that, there was no significant difference in PXRD patterns between raw material and the residual phases after the solubility experiments for PRO and all cocryatals (Appendix A).

### 2.5. Nitrogen Heterocyclic Ring CCFs That Cannot Form Cocrystals with PRO

Under ethyl acetate-assisted grinding conditions, some nitrogen heterocyclic ring CCFs could not form cocrystals with PRO, such as heterocyclic amino acid and purine/pyrimidine (Table 4). There were probably two reasons. One was the hydrogen bound to the CCF too tightly; the other was the CCFs themselves being prone to forming intramolecular/intermolecular hydrogen bonds. As a result, they failed to form cocrystals with PRO.

### 2.6. Substituted Pyridine Derivatives

Pyridine could not form cocrystal with PRO. As shown in Table 5 and Table 6, in contrast to the carboxyl group, amide group and amino group were good substituents, which meant the electron inductive effect showed a very strong effect in cocrystallization of PRO and pyrazine derivatives. Single crystal data could be obtained when there was a p-NH_2_ or p-CONH_2_ on pyridine ring. The cocrystal of o-CONH_2_ on pyridine ring (PA) with PRO could be confirmed by PXRD, TG, DSC and ^1^H-NMR, while the single crystal could not be obtained successfully.

### 2.7. Substituted Pyrazine Derivatives

Pyrazine could not form cocrystal with PRO. Similar to pyridine derivatives, the cocrystallization could not proceed between PRO and piperazinecarboxylic acid. The reaction could take place between PRO and piperazine with amide or amino-substituted (Table 7). The results of PRO-PZA were similar to PRO-PA, cocrystallization took place, but the single crystal was difficult to obtain. Cocrystallization of PRO and piperazine with methyl-substituted was negative because there was no active hydrogen in the molecule.

## 3. Discussion

The crystal formation of PRO cocrystals included two processes: nucleation and growth. In the nucleation stage, the clusters generated by weak interactions between API and CCF generally retained the essential structural characteristics of cocrystals. The cocrystal molecules were in an ordered arrangement with the assistance of electron-donating groups to precipitate crystals with suitable sizes for single-crystal testing. However, the electron-withdrawing group was unable to arrange the cocrystal molecules in order, or the intermolecular force between cocrystal molecules was too weak to form appropriate unit cells, resulting in the precipitation of thin/small/hardened cocrystal particles, which were unsuitable for SXRD. The reason for not obtaining a single crystal was explained by calculation. The initial structure was constructed according to the electrostatic potential and synthetic subrules for cocrystal formation of PRO and PA. MTD-GC (metadynamics-genetic crossing) was used to search and optimize various cluster structures, followed by sorting their stability with the GFN0-xTB method [19]. The solvent effect of ethyl acetate was calculated by the GBSA model. The above work was performed in the CREST [20] and xTB software [21].

In general, dimers (1API/1CCF) were stable for the cocrystal, which could obtain single-crystal data. The calculation results showed that the tetramers (2PRO/2PA) were stable, with a stable hexagonal cyclic hydrogen bond formed between the amide groups of two adjacent PA. Two molecules of PRO were stabilized with adjacent PA through the Van der Waals force (Figure 5). The thin lines reflected the various cluster structure and the thick lines were the most stable. Figure 6 shows the tetramer structure in ethyl acetate as the solvent crystallized experimentally. The structural fluctuations of the cluster in ethyl acetate solvent were increased and led to a deficiency in order structure, which might be one of the reasons why the PRO-PA crystal was difficult to grow in ethyl acetate. Dissimilarly, the CIF file of PRO-INA showed that one molecule of PRO was hydrogen-bonded to two molecules of INA, and there was no interaction between adjacent PRO molecules (Figure 7).

## 4. Materials and Methods

### 4.1. Materials

PRO (purity 99.4%) was obtained from Zhejiang Xianju Pharmaceutical Co., Ltd. (Xianju, China, Lot number: Y011-200507). CCFs were purchased from Shanghai Bide Pharmatech Co., Ltd. (Shanghai, China). All reagents and chemicals were commercially available and used directly. 

### 4.2. Preparation of the Cocrystals and Single Crystals

All PRO cocrystals were prepared by ethyl acetate-assisted grinding in a ball mill (QM3SP2L, Nanjing Chishun Science & Technology Co., Ltd., Nanjing, China). The grinding experiments were performed by the addition of an equimolar of PRO (943.4 mg, 3 mmol), corresponding CCFs (3 mmol) and 0.1 mL ethyl acetate to a 250 mL agate grinding jar. The mixture was then ground at a frequency of 28 Hz for 40 min [22].

An appropriate amount of the PRO cocrystals was dissolved in ethyl acetate. The solvent was slowly volatilized, and the obtained crystals were analyzed by single-crystal X-ray diffraction.

### 4.3. General Methods 

Powder X-ray diffraction (PXRD) patterns were recorded with a BRUKER D8 advance diffractometer system with CuKα1 radiation (λ = 1.5406 Å, 40 kV, 40 mA) over the interval 3–60°/2*θ*. Thermo gravimetric analysis-differential scanning calorimetry (TG-DSC) was conducted on TGA/DSC^3+^ equipment under a flow of nitrogen (20 mL/min) at a scan rate of 10 °C/min from 40 to 400 °C. Fourier transform infrared spectroscopy (FT-IR) was performed with a Bruker EQUINOX 55 FT-IR spectrometer (Billerica, MA, USA). A total of 64 scans were collected over a range of 4000–400 cm^−1^ with a resolution of 0.2 cm^−1^ for each sample. A Jeol JSM-6100 scanning electron microscope (SEM, Akishima, Japan) was used to obtain photomicrographs. Samples were mounted on a metal stub with adhesive tape and coated under a vacuum with platinum. Nuclear magnetic resonance (^1^H-NMR) was recorded using a Bruker 400 MHz instrument using DMSO-*d*^6^ as a solvent and TMS as an internal standard. Single crystal X-ray diffraction (SXRD) data were collected by Rigaku AFC-10/Saturn 724-CCD diffractometer (Tokyo, Japan) equipped with a graphite-monochromatized MoKa radiation (0.71073 Å) up to a 2 h limit of 50.0° at room temperature (25 °C).

### 4.4. Stability Study

For the stability study, the powder samples of 100–200 mg cocrystals in 5 mL uncapped glass vials were placed in a stability chamber (Bluepard Yiheng, Shanghai, China) at 60 ± 2 °C, 90 ± 5% RH and light exposure of 4500 ± 500 lx for 10 days, respectively. Then, the samples were analyzed by PXRD for 5 and 10 days.

### 4.5. In Vitro Dissolution Tests

The in vitro dissolution tests was carried out according to the guideline and procedure specified in Chinese Pharmacopeia. The dissolution of the experiment was performed on the dissolution apparatus (RC806D, Tiandatianfa, Tianjin, China) at 37 °C with the rotation speed set at 100 rpm, and the samples were taken at 5, 15, 30, 60, 120, 240, 300 and 360 min. The samples were filtered through 0.45 μm membrane filters and measured on HPLC (LC-20A, Japanese Shimadzu Corporation, Kyoto, Japan) coupled with a diode array detector. The wavelength was 254 nm, and the chromatographic column was Inertsil ODS-3 C18 (5 μm × 4.6 mm × 150 mm). The mobile phase was methanol/water (65:35, *v*:*v*). The flow rate of the mobile phase was 1 mL/min, and the injection volume was 20 μL.

## 5. Conclusions and Outlook

In the present work, six PRO cocrystals were prepared and evaluated in stability and solubility, of which four cocrystals had single crystal data. All cocrystals were prepared by grinding in a ball mill without solvent. They were more soluble than PRO. The exploration of more CCFs is still underway. 

Based on the cases studied in this paper, the empirical rules of PRO-CCFs were summarized according to the electronic effects of the aromatic substituent. Such rules could be used to rationalize the possibility of a combination of PRO with various CCFs, but there was no guarantee of working in other cases. The specific CCF of formed or failed cocrystals is a very complex process driven by many factors. At present, choosing CCFs relies largely on experience. Meanwhile, several strategies and methods have been developed to aid in predicting the possibility of CCF cocrystal formation.

Among these approaches, data-driven machine learning (ML) methods can provide robust mathematical models to predict the CCFs selection, improved by good quality and amount of data from statistical perspectives. In this work, a reliable cocrystal dataset for PRO-CCFs was obtained by collecting successful and failed samples. The failed cases were particularly useful, which was not readily available to be found in published papers.

Benefiting from our rich data and experimental practices, we are building ML-based classifiers to predict PRO-CCFs cocrystal formation. Selected molecular descriptors can be acted as “input” while a successful cocrystal or not as “output”. The former involves molecular size, flexibility, Hansen solubility parameters, hydrogen bond tendency, etc. The predicted models will be trained by random forest (RF), support vector machine (SVM) and artificial neural network (ANN) algorithms. We expect such models to become useful tools for the design of cocrystal.

## Figures and Tables

**Figure 1 molecules-28-04242-f001:**
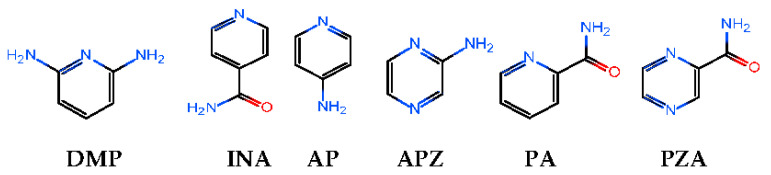
Chemical structures of the CCFs used in this study.

**Figure 2 molecules-28-04242-f002:**
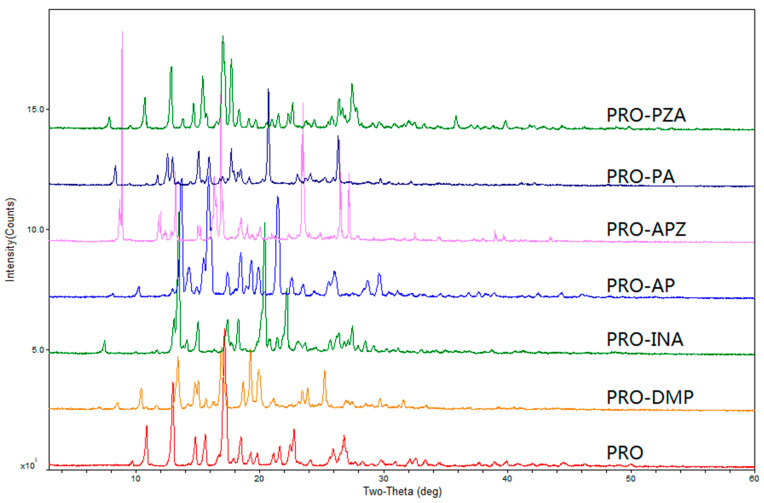
Powder X-ray diffraction (PXRD) of PRO and 6 cocrystals.

**Figure 3 molecules-28-04242-f003:**
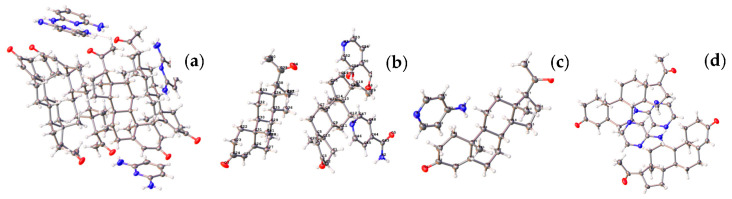
Structure of (**a**) PRO-DMP, (**b**) PRO-INA, (**c**) PRO-AP and (**d**) PRO-APZ.

**Figure 4 molecules-28-04242-f004:**
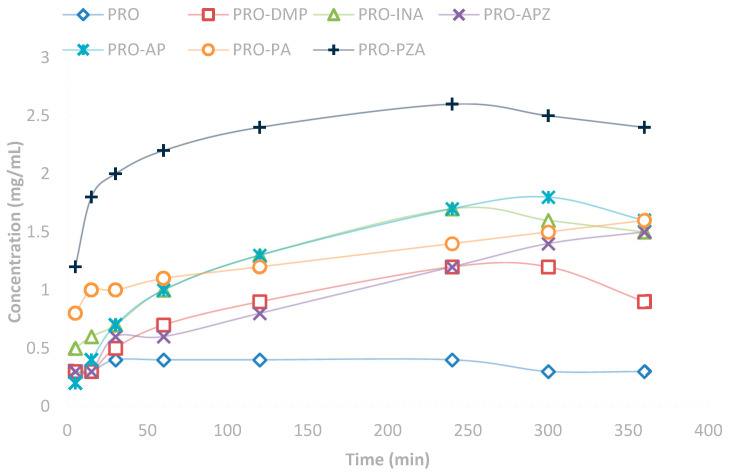
In vitro dissolution profiles for commercial PRO and 6 cocrystals in water.

**Figure 5 molecules-28-04242-f005:**
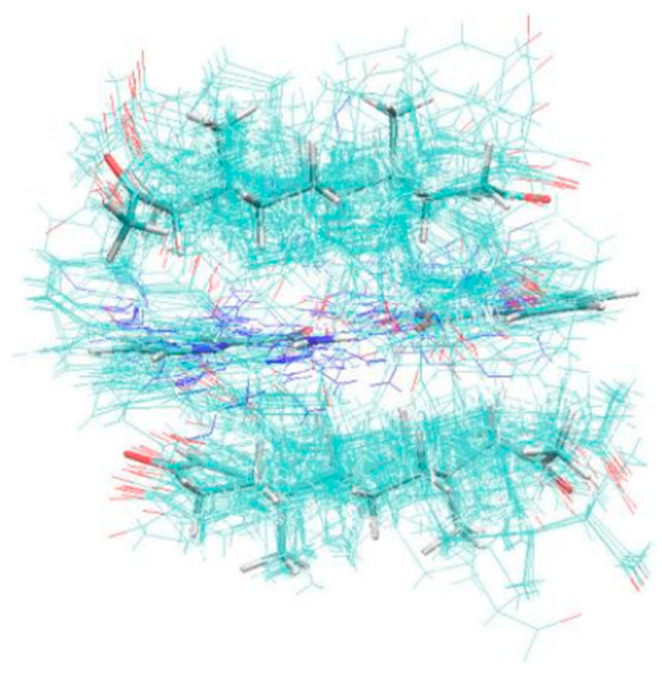
Polymer diagram of 2PRO-2PA.

**Figure 6 molecules-28-04242-f006:**
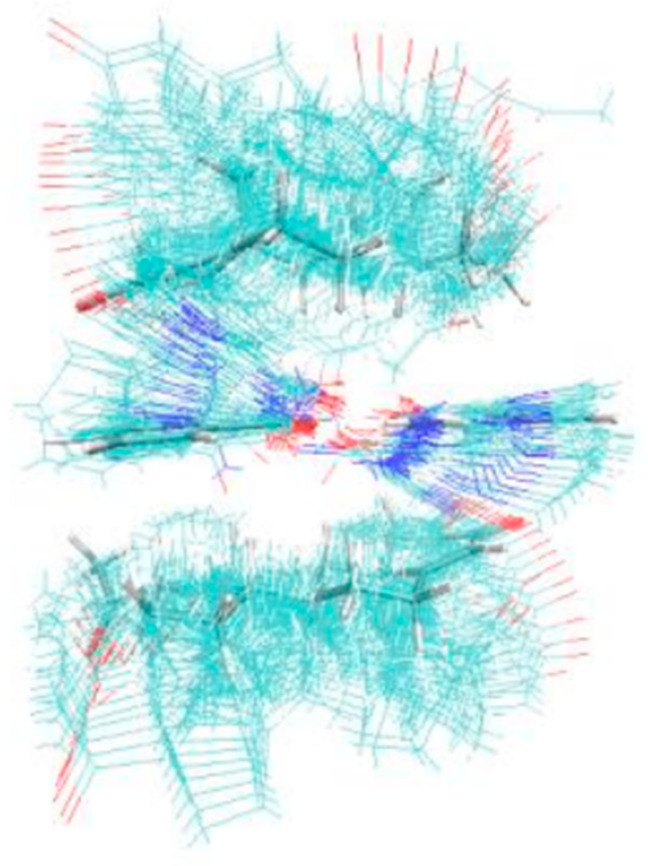
Polymer diagram of 2PRO-2PA in ethyl acetate.

**Figure 7 molecules-28-04242-f007:**
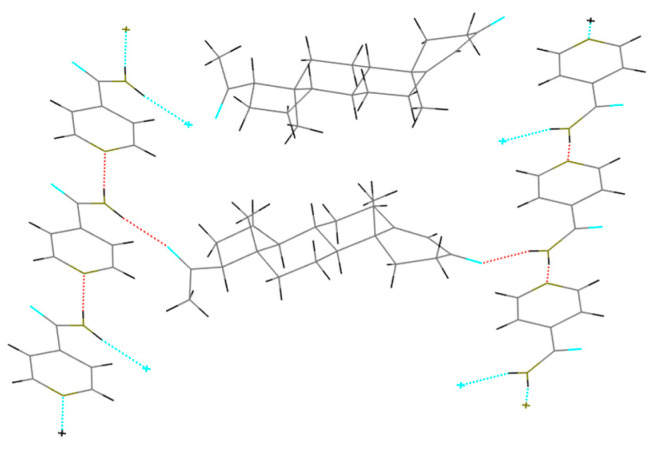
Polymer diagram of CIF file for PRO-INA.

**Table 1 molecules-28-04242-t001:** Brief introduction to Progesterone.

Parameters	Data
Chemical name	4-Pregnene-3,20-dione
Chemical formula	C_21_H_30_O_22_
Molecular weight	314.46 g·mol^−1^
CAS Registry No.	57-83-0
Chemical structure	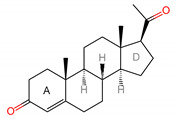

**Table 2 molecules-28-04242-t002:** Melting points PRO and 6 cocrystals.

Compounds	Melting Points (°C)
PRO	132
PRO-DMP	110
PRO-INA	130
PRO-APZ	112
PRO-AP	122
PRO-PA	122
PRO-PZA	126

**Table 3 molecules-28-04242-t003:** Crystallographic parameters of PRO-DMP, INA, AP, APZ.

Compounds	PRO-DMP	PRO-INA	PRO-AP	PRO-APZ
CCDC no.	2,131,361	2,131,362	2,131,375	2,131,374
Empirical formula	C_26_H_37_N_3_O_2_	C_27_H_36_N_2_O_3_	C_26_H_36_N_2_O_2_	C_25_H_35_N_3_O_2_
Formula weight	423.59	436.58	408.57	409.56
Temperature / K	116.10 (14)	116.40 (14)	113.10 (14)	113.35 (10)
Crystal system	monoclinic	orthorhombic	orthorhombic	triclinic
Space group	P2_1_	P2_1_2_1_2_1_	P2_1_2_1_2_1_	P_1_
a / Å	13.0819 (5)	7.8943 (5)	7.6234 (6)	10.4819 (10)
b / Å	18.6192 (6)	12.9909 (17)	12.2741 (8)	10.7971 (10)
c / Å	19.7511 (8)	46.685 (2)	24.4042 (16)	11.1480 (11)
α/°	90.00	90.00	90.00	113.753 (9)
β/°	107.375 (4)	90.00	90.00	99.185 (8)
γ/°	90.00	90.00	90.00	90.928 (8)
Volume / Å^3^	4591.3 (3)	4787.7 (7)	2283.5 (3)	1135.45 (19)
Z	8	8	4	2
ρ_calc_ / mg mm^−3^	1.226	1.211	1.188	1.198
*μ */ mm^−1^	0.078	0.078	0.075	0.076
F (000)	1840	1888	888	444
Crystal size / mm^3^	0.40 × 0.35 × 0.33	0.40 × 0.40 × 0.31	0.22 × 0.21 × 0.21	0.45 × 0.43 × 0.42

**Table 4 molecules-28-04242-t004:** Some nitrogen heterocyclic ring CCFs could not form cocrystals with PRO.

Category	Specific Compounds
Heterocyclic amino acid	L-histidine, L-proline, DL-tryptophan
Purine/pyrimidine	adenine, cytosine, guanine, thymine, uracil
Heterocyclic compound	2-acetylthiazole, 2,3′-bipyridine, 4,4′-bipyridine, folic acid, 5-(2-hydroxyethyl)-4-methylthiazole, imidazole, orotic acid, piperazine, pyrazine, pyridine, saccharin

**Table 5 molecules-28-04242-t005:** Cocrystallization between PRO and pyridine with one substituent.

Pyridine with One Substituent	*o*	*m*	*p*
-COOH	−	−	−
-CONH_2_	+ (PA)	−	+ √ (INA)
-NH_2_	−	−	+ √ (AP)

+: positive cocrystallization; −: negative cocrystallization; √: single crystal.

**Table 6 molecules-28-04242-t006:** Cocrystallization between **PRO** and pyridine with two substituents.

Pyridine with Two Substituents	Position	Reaction
-COOH	2.6	−
-NH_2_	2.6	+ √ (DMP)

+: positive cocrystallization; −: negative cocrystallization; √: single crystal.

**Table 7 molecules-28-04242-t007:** Cocrystallization between PRO and substituted pyrazine.

Substituted Pyrazine	Reaction
-COOH	−
-CONH_2_	+ (PZA)
-NH_2_	+ √ (APZ)
-CH_3_	−
-CH_3_, -CH_3_, -CH_3_	−
-CH_3_, -CH_3_, -CH_3_, -CH_3_	−

+: positive cocrystallization; −: negative cocrystallization; √: single crystal.

## Data Availability

The data presented in this study are available in Appendix A.

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
