# Peer review of "Cocrystallization of Progesterone with Nitrogen Heterocyclic Compounds: Synthesis, Characterization, Calculation and Property Evaluation"

_molecules, 2023, doi:10.3390/molecules28104242_

Round 1
Reviewer 1 Report
Manuscript presented by Juan Xu et al. shows a study about synthesis development new dosage forms and delivery methods of progesterone. For this purpose, the Authors designed and prepared 6 progesterone cocrystals with nitrogen heterocyclic compounds. Additionally, the Authors used metadynamics-genetic crossing method. The topic is very important due to future solution of dosage of progesterone. Thanks to The Authors for conducting this intriguing study.
An already well written and prepared manuscript. Easy to read and follow. Some aspects should be improved. I recommend the article to publish in Molecules but first the paper should be corrected. My decision – reconsider after minor revision. Comments to be considered, in order to further improve the manuscript quality:
(1) Please add “Metadynamics-genetic crossing” to keywords section.
(2) Introduction: Please present data about PRO in table with chemical structure (which is include in scheme 1). It is easier to follow of text.
(3) Would you explicitly specify the novelty of your work? What progress against the most recent state-of-the-art similar studies was made?
(4) Please check if all Figures, Schemes and Tables are cited in main text. Avoid lumping the data from Tables (eg Tables 3-6). Instead summarise the main contribution of each Tables in a separate sentences.
(5) Please add compound numbering. Consider adding bold text for the abbreviations of PRO and its cocrystals.
(6) Please reorganise the text and connect it with Figures, Schemes and Tables (see other Molecules articles).
(7) Scheme 1 – it is Figure not reaction scheme. Please correct.
(8) Please describe the safety of obtained co-crystals as potential medicines.
(9) Preparation of the cocrystals and single crystals – if this procedure is common (if yes please add reference) or new (if it is novel method please underline this fact in the manuscript)?
(10) Superscripts and subscripts as well as commas and dots should be checked and correct. Avoid extra or lack of spaces and enters (eg Scheme 1). Pay special attention of reference section. Correct in whole manuscript.
(11) Please check the reference. There are typographical errors in some positions (eg. dash type). Correct it.
(12) The English should be improved. Please use British English in the manuscript.
The English should be improved. Please use British English in the manuscript.
Reviewer 2 Report
This work presents an incremental, yet valuable contribution to developing an empirical approach to co-crystallizing PROG with amine heterocyclics for improving pharmaceutical delivery. The work is ready for publication, except for a few minor clarifications. The co-crystals with AP and APZ the authors report a melting transition, but from their DSC in the SI, it appears an exothermic transition is occurring in this range which contradicts melting. Finally, the NMR explanation should provide more detail in regards to specific integration in the results section.
